# "Four Joints of Power" Innovation of Community Involvement in Medical Waste Management of Bed-Bound Patients in Thailand

**Sakchai Pattra [1], Cung Nawl Thawng [2] and Sanhawat Chaiwong [3],\***

1   The Faculty of Arts and Science, Chaiyaphum Rajabhat University, Chaiyaphum 36000, Thailand
2   Department of Biology, New Mexico State University, Las Cruces, NM 88003, USA
3   School of Public Health, Walailak University, Nakhon Si Thammarat 80160, Thailand
\*   Correspondence: sanhawat.ch@wu.ac.th

**Abstract:** This study aims to encourage innovative participation in the management of medical waste by bedridden patients in the research region of Khon Sawan, Chaiaphum Province, through research and development. The steps were as follows: Phase 1: Study of bedridden patient waste management situations using the amount of waste generated through innovation with relatives, non-relatives, village health volunteers (VHVs), and community leaders. Phase 2: Developing creative waste management engagement requires two steps: (1) analyzing the problem or its cause and generating management alternatives through collaborative brainstorming with a community member and (2) gathering the thoughts and suggestions of a number of agency specialists. The outcome is a novel model of participation in waste management by bedridden patients termed "Four Joins of Power," which includes (1) participatory activities and enhancing community knowledge and attitudes, and (2) providing information on the management of each type of waste. (3) cooperation in waste management (analytical thinking, planning, execution, etc.) and regulation by mutually agreed-upon rules. (4) joint expansion of the waste management network: Phase 3 is the innovation trial, and Phase 4 is the innovation assessment. The paired t-test was used to compare pre-and post-development knowledge and attitudes, and to conduct qualitative data analysis. In Phase 3, after implementing collaborative innovations, the average knowledge ($\overline{X}$ = 13.23) and attitudes ($\overline{X}$ = 4.14) regarding waste management increased considerably ($p < 0.05$), and in Phase 4, waste management behavior comprising sorting, storage, and disposal was observed. There were progressively substantial gains ($\overline{X}$ = 4.25 and $\overline{X}$ = 4.27). Among the most collaborative participants, 93.50% were satisfied. To reduce the amount of waste that must be sorted and collected, it is necessary to emphasize the participation of people and networks from all sectors in the area through joint thinking, planning, and comprehensive analysis, to ensure the sustainability of waste management in the community.

**Keywords:** innovation; medical waste management; participation; bedridden patients; Thailand

## 1. Introduction

Medical waste is described as solid or semi-solid waste created throughout the procedure as well as any waste generated during medical treatment and disease prevention. It generates a variety of potentially hazardous compounds known as "medical waste [1,2]. Infectious waste such as sharps and communicable diseases such as AIDS, hepatitis B, and hepatitis C account for 10% to 25% of medical waste, whereas non-infectious waste accounts for 75% to 95% [3–5]. Owing to the harmful yet contagious nature of medical waste, it is one of the particular issues of municipal solid waste that is of utmost significance [6]. It is not only required by law to manage medical waste effectively and securely but it is also a social responsibility because this process affects the health and safety of numerous employees, patients, and other individuals [7]. It is crucial to consider how to collect, separate, transport, store, dispose of, and treat medical waste to safeguard the environment

and ensure public safety [5]. Thus, the improper treatment of medical waste has negative environmental effects [8,9]. It is estimated that 7–10 billion tons of waste are produced annually on a global scale, of which only 2 billion tons are municipal solid waste, to which medical waste contributes a negligible amount [10]. According to estimates, approximately 80% of medical waste in developing countries is mixed with regular waste. Approximately 10–25% of biological waste is hazardous, whereas 75–95% is harmless [10,11]. It is vital to note that each type of hazardous waste is unique in terms of its ingredients, sources, and danger potential. The subcategories of hazardous waste include infectious, pathological, needle, pharmaceutical, clinical, and radioactive waste [12]. Medical waste is inevitably produced by the provision of preventive, promotive, and curative services. Incorrect waste handling of waste may have devastating effects on human and animal health and ecosystems [13]. It has been established that this waste may be swept into rivers, streams, and groundwater, which may lead to outbreaks of waterborne illnesses and the spread of several drug-resistant species. Due to inappropriate medical waste disposal, direct contact with contaminated waste containing sharp edges, such as needles and shattered vials, potentially presents a health risk, causes skin damage, and spreads pathogenic microorganisms [12]. In most communities, waste management is a serious challenge, particularly in developing nations that do not have environmental education campaigns [10,14,15]. Risk waste consists of numerous components of hospitals and other biomedical wastes deemed hazardous to the public because of the existence of specific components with an inherent capacity to spread illness or damage to populations in underdeveloped nations [16].

Based on what occurred to infectious waste in Thailand between 2013 and 2016, the Office of Natural Resources and Environmental Policy and Planning reported that the amount of infectious waste in 2017 was between 50,481, 52,147, 53,868, and 55,646 tonnes, respectively, and that 0.41 kg of waste was generated daily in Thailand [17–20]. One contributor to the increase in medical waste is bedridden patients, particularly the predicted increase in the number of elderly people in Thailand. By 2030, there are 152,749 bedridden groups. The predicted inaccuracy was 1.75%. The number of bedridden patients fluctuated between 150,249 and 155,596 based on a parameter estimate incorporating a 20% increase in transition probability. The number of bedridden patients will increase to 311,256 by 2037, an increase of between 8000 and 10,000 per year, contributing to the expansion of medical waste in Thailand [21]. To avoid the spread of illness and medical waste-related mortality, it is vital to properly manage social and economic factors that are primarily attributable to the rapidly increasing amount of medical waste in the community. Along with the increase in waste from immobile patients, there has also been an increase in infected wounds in official healthcare institutions, owing to how doctors handle wounds and other developments. There are other factors for proposing that bedridden patients receive care at home, which is consistent with the requests of patients and their families for accessible care products.

Insufficient grasp of management, inadequate practices, and a high frequency of sharp-object injuries on the job was identified. Laboratory examinations of dustmen revealed lung tuberculosis in 3.4%, parasites and intestinal infections in 5.1%, and Hepatitis B Surface Antigen (HBsAg) positivity in 8.5% [20]. Medical waste management consists of waste separation, storage, transportation, and disposal, which require the competence and cooperation of all concerned parties and the provision of appropriate equipment and finances [22]. Waste disposal is a serious challenge in most communities, particularly in developing nations without environmental education initiatives [12]. Public healthcare facilities, such as hospitals, are increasingly providing in-home care for a variety of diseases and conditions with physicians and health officials [23]. The majority of activities involving bedridden patients include daily care, which is a source of infectious waste, such as gauze and gloves. It was discovered that inappropriate disposal of infected and noninfectious waste that has been in contact with a patient can culminate in disease transmission [24]. The research staff was interested in investigating the development of innovations for bedridden patients' engagement in waste management to use the study's findings as a reference for addressing pertinent problem scenarios and adopting innovations. which has



been elongated. There has been a shift in patients' and relatives' attitudes and caregivers, including community members, toward avoiding the spread of infectious waste from immobile patients for the sake of their health and that of their families.

The objectives were to investigate the current state of waste disposal for bedridden patients, create ideas, and evaluate bedridden patients' engagement in waste management. In the future, the invention should be used as a guide to involving community members in the waste management of bedridden patients so that it is effective and safe for the community. The research and development approach was implemented by dividing the research into four phases. Phase 1: A study of bedridden patient waste management situations. Phase 2: Developing creative waste management engagement for bedridden patients: Phase 3 was the trial of the invention and Phase 4 was the evaluation of the innovation. The findings of this study will result in sustainable innovation in infectious waste management that can be used to manage infectious waste in bedridden patients, reduce infectious waste in the future without affecting people's health or the environment, and avoid placing a burden on the government sector to manage the future waste management budget.

## 2. Literature Review

As the global population swells and the demand for medical services increases, medical waste management becomes one of the many complex and challenging societies that must be confronted. The World Health Organization (WHO) defines medical waste as "waste created during the diagnosis, treatment, or immunization of humans or animals". Medical waste that is not properly managed and disposed of poses a significant risk of illness or harm to healthcare workers as well as a lower risk to the general public through the transmission of microbes from healthcare facilities to the environment [25,26]. The disposal of medical waste is a problem of great scope. The United States generates more than 3.5 million tonnes of medical waste annually at an average cost of $790 per ton [27]. As the access to medical services improves and more people can obtain contemporary medical treatment, the amount of medical waste produced in developing countries is escalating significantly. The shift away from multi-use medical devices toward safer, single-use medical gadgets contributes to the accumulation of medical waste in poor countries. These combined trends are producing rapid growth in the amount of medical waste in emerging nations that require safe disposal [28]. In the industrialized world, a rapidly aging population is the primary driver of increased medical system utilization, and this rising medical system utilization is resulting in an increase in medical waste output [29]. This article outlines the challenges associated with the disposal of medical waste. First, the nature and origin of medical waste in many regions of the world are investigated. This is followed by a review of both mandatory laws and guidelines for handling medical waste in these places. Next, existing techniques for managing medical waste are detailed, with an emphasis on in-facility collection, separation, transportation, and disposal methods. Alternative treatment options and the necessity of reducing the proportion of non-infected medical waste in the infectious medical waste stream will then be examined. Finally, recommendations for improvement measures, comprising improved education for healthcare personnel and standardization of waste receptacles throughout the facility, will be made. This review demonstrates that the amount of infectious waste produced and the accompanying harm can be decreased by enhancing waste sorting at the point of disposal, standardizing waste disposal streams, and properly educating the healthcare staff. Four terminologies are commonly used interchangeably when discussing medical waste, but none has a broadly agreed definition [30]. Hospitals and medical, regulated, and infectious medical waste were included in this study. To offer clarity and consistency throughout this study, the term "medical waste" will be used to refer to all waste generated at any hospital or healthcare-related facility, consistent with the United States Environmental Protection Agency's definition of medical waste [31]. The phrase "infectious medical waste" refers to a subset of healthcare facility waste that cannot be disposed of in municipal solid waste systems owing to pathogen concerns [32,33]. "Hospital waste" (or "solid waste") refers to

any disposed of waste, biological or nonbiological, that is not intended for reuse. The term 'medical waste' refers to materials resulting from the diagnosis, treatment, or immunization of human or animal patients.

The term "infectious waste" refers to medical waste that may transmit infectious illnesses. In the Medical Waste Tracking Act (MWTA), Congress and the EPA employ the phrase "controlled medical waste" as opposed to "infectious waste", because of the distant probability of disease transmission. Thus, "medical waste" is a subset of "hospital waste", and "regulated medical waste", which, from a regulatory standpoint, is synonymous with "infectious waste", is a subset of "medical waste". Infectious wastes have the potential to cause diseases. This concept necessitates an examination of the factors necessary for the induction of disease, including dose, host vulnerability, presence of a pathogen, the virulence of a pathogen, and the most commonly lacking factor, an entry point. For waste to be infectious, it must contain pathogens with sufficient virulence and quantity to expose a susceptible host to waste, resulting in the development of an infectious disease [25].

Techniques for managing medical waste include source control and separation, collection for storage and transit, treatment, and disposal. Disease propagation in the environment is affected by the difficulties associated with each of these processes. Various phases of medical waste management potentially result in the spread of organisms that are harmful to public health [32]. The lack of medical waste disposal procedures in Thai hospitals has aggravated the medical waste problem. For instance, waste collectors frequently have respiratory illnesses and unintentional accidents from sharp items. The lack of regulated procedures for the disposal of medical waste in hospitals in Thailand has exacerbated this problem of medical waste. For most hospitals, the Ministry of Public Health erected incinerators for medical waste. However, there are still many operational issues, such as a lack of standards for incinerators, personnel with knowledge of incinerator operations, and inefficient maintenance of incinerators, as well as health and environmental issues resulting from pollutants and poisoning caused by the burning of infectious waste in incinerators. Mismanagement of infectious waste results in environmental pollution and unpleasant odors due to harmful pathogens that may lead to infections such as typhoid, cholera, tuberculosis, and other diseases such as hepatitis and HIV/AIDS [33]. Even if the amount of infectious waste produced is negligible in proportion to the overall medical waste, this waste will arise from waste management techniques that are not in compliance with the hygienic principles of infectious waste management. When combined with other debris, it can constitute infectious waste [34]. In addition, some hospitals prefer to dispose of some types of infectious waste, while the remainder is submitted to the local government for disposal. Under the Ministry of Public Health, 62.43% of staff dispose of infectious waste themselves, while 23.29% transfer the remaining 14.22% to private businesses. Hospitals employ this disposal approach by delivering waste to the private sector and adjacent hospitals with waste incinerators [35]. Therefore, it is essential to educate caregivers on the correct treatment of infectious waste created by bedridden patients in order to decrease expenses and prevent adverse effects on public health and the environment.

## 3. Materials and Analysis

Figure 1 provides an overview of the stages involved in the process of developing innovation for activities in phases 1–4. This is done to encourage new ideas named "4 Joints of Power".

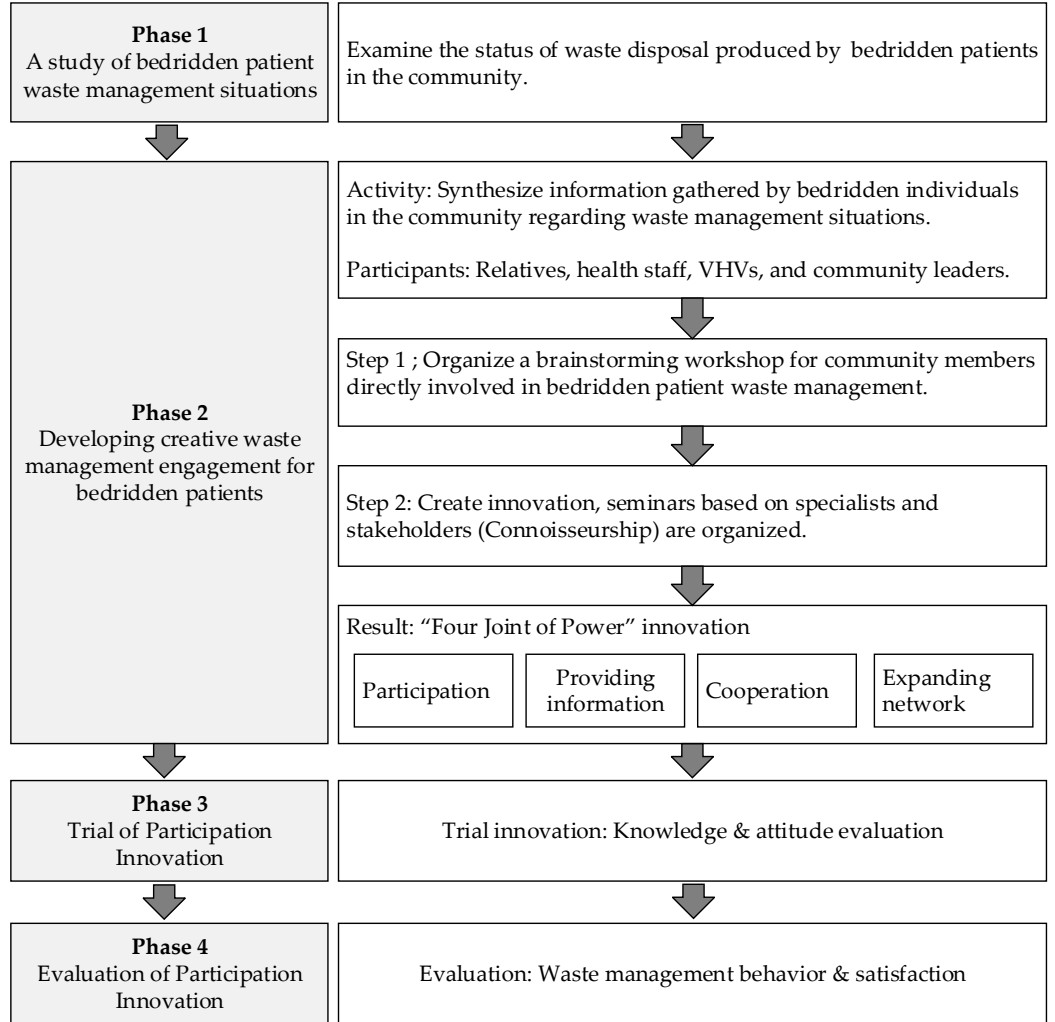

**Figure 1.** Innovation development process.

### 3.1. The Scope of the Analysis

Participatory action research: PAR using the Deming cycle (Plan-Do-Check-Act) and participatory learning: PL to develop with the participation (social engagement) of community members in problem-solving and infectious waste management in bedridden patients with empirical and qualitative data collection and content analysis from interviews with key informants to categorize content, synthesize them as joint issues, and summarize data issues by analytic induction, including returning the data to stakeholders and the community to verify the accuracy of the data again before dissemination of the research results. Therefore, "4 Joints of Power" is an invention that may be utilized for the administration of community medical waste to bedridden individuals.

### 3.2. Step of the Research Process

Phase 1: A study of bedridden patient waste management situations.

Documentary research and published theories on waste management were analyzed to develop a preliminary research framework for field data collection. In Chaiyaphum Province. The quantitative data included general data and the amount of waste generated by bedridden patients, whereas the qualitative data included the circumstances of bedridden patients and waste management in the community. The population was separated into two groups: 78 relatives or caregivers of bedridden patients and 110 VHVs in the Khon Sawan region of Chaiyaphum Province from November 2020 to May 2021.

They were tasked with gathering information regarding the waste status of the bedridden patients. Developing innovative collaboration from immobile persons in waste management knowledge, attitude, and behavior of representatives who are relatives or caregivers, compared before and after the innovation's implementation.

The research instrument was a three-part semi-structured interview.

Part 1: General data.

Part 2: Quantity data of waste generated by bedridden patients (in one week).

Part 3: Bedridden patient condition and waste management in the community. Five inquiries were conducted.

(1) The current situation of the waste problem caused by immobile people and an assessment of the community's future condition.
(2) The criteria for the creation of community waste management for bedridden patients.
(3) Family members of bedridden patients, non-relative caregivers, and VHVs.
(4) The knowledge and attitudes of waste management among bedridden patients in the community.
(5) The behavior of waste management among bedridden patients at home. Each participant was interviewed for no longer than forty minutes.

The qualitative analysis employs content analysis to categorize material into broad concerns, expound on the content, and triangulate the data to ensure correctness with relevant groups before reaching comprehensive and valid findings. Mean and frequency were then analyzed descriptively.

Phase 2: Developing creative waste management engagement for bedridden patients.

This is a two-step process for studying a problem in which the source of the problem and ways to control waste from bedridden people are considered, and innovations for the participation of bedridden people in waste management are developed from May 2021 to September 2021.

Step 1: The root cause analysis and waste management plan for immobile patients constituted the analysis and synthesis of medical waste management problems produced by bedridden patients, as well as the types of disposal methods. Listen to the viewpoints of those involved in caring for bedridden patients in the community and those interested in waste management in the community, such as family, caregivers, health professionals, VHVs, community leaders, and local government organizations, to offer everyone a chance to voice their views on medical waste problems and management solutions. We then provide a summary of the medical waste management recommendations. Three local administrative organizations, five heads of government agencies, and six community leaders discussed additional thoughts and proposals as guidelines for waste management of bedridden patients throughout the conference. In the second stage, the medical waste management recommendations were presented to the community for additional brainstorming with the target population in mind so that they might be the primary players in carrying out their responsibilities, family members of bedridden patients, caregivers, and VHVs with at least one year of experience in caring for bedridden patients handling medical waste in the community.

Step 2: Organize a forum composed of specialists and connoisseurs to provide innovative waste management with the participation of bedridden patients. The researcher chose to use a purposeful selection method with 15 participants, including five caregivers with at least three years of experience in caring for bedridden patients, five academics from public health authorities, three academics with knowledge and experience in infectious waste handling, and two professors from Chaiyaphum Rajabhat University. After the group discussion, the research team presented the participants with four questions on how to properly handle the waste of bedridden patients. The following is a list of questions that were asked: (1) the waste situation caused by bedridden patients, (2) the waste management caused by bedridden patients, (3) the participation of the community in bedside waste management, and (4) novel recommendations for the participation of bedridden patients in

waste management that could be used as a guide for the participation of bedridden patients in further data collection.

After the forums, the guidelines for documenting both the discussion in the focus group and the meeting to develop recommendations for creating inventive ways for individuals who are bedridden to participate in waste management. Extracting and encoding the data, followed by classification, investigation, and synthesis, are required to complete the content analysis.

In conclusion, it was proposed that technological solutions should be created to assist people who are unable to move in the disposal of their waste. Then, activities would be utilized to minimize the amount of infectious waste in the community, as determined by the analysis of data in the step called "4 Joints of Power" as follows: (1) participatory activities and enhancing community knowledge and attitudes, and (2) providing information on the management of each type of waste. (3) cooperation in waste management (analytical thinking, planning, execution, etc.) and regulation by mutually agreed-upon rules. (4) joint expansion of the waste management network: In Phase 3, these technologies are a trial of Participation Innovation.

Phase 3: Trial of Participation Innovation.

The concept was evaluated from September 2021 to January 2022, in the Khok Mang Ngoi sub-district of Khon Sawan by conducting activities to increase knowledge of innovation and participation in waste management among bedridden patients' relatives, non-relative caregivers, VHVs in the sub-district, and speakers from the Chaiaphum Provincial Health Department and Environmental Health deliver information on awareness and innovation resulting from Step 2 of Phase 2 to develop the target audience's attitudes toward waste management. All sample sizes were engaged in waste management before and after the activities. Sample groups who volunteered for a forty-person research project included relatives, non-relative caregivers, and VHVs at Khok Mang Ngoi Health Promoting Hospital.

The instrument for measuring general and medical waste knowledge was a questionnaire. Thailand's Ban Khao District Chaiyaphum was home to thirty volunteers. Using the Kuder-Richardson coefficient (KR20) and the Cronbach alpha coefficient model G, which were 0.89 and 0.81, respectively, the confidence level of the questionnaire was evaluated.

Descriptive statistics such as frequency, percentage, mean, and standard deviation were used, and analytical statistics such as a t-test were used to compare individuals' knowledge and attitudes about medical waste management before and after at a statistical significance level of.05.

Phase 4: Evaluation of Participation Innovation.

The 4th phase began in January 2022 and end in March 2022. This was a follow-up evaluation after the new idea had been used for six months in the Khok Mang Ngoi sub-district community in Khon Sawan District by a group involved in the management of the community. It consisted of an assessment of the behavior and satisfaction of 65 relatives and caregivers of bedridden patients in the Khok Mang Ngoi sub-district. Research tools include quantitative assessments, such as waste management behaviors and satisfaction, which were assessed using key formats. Before utilizing the questionnaire to test a group, it was tested in the Ban Khao district, Chaiyaphum province, where 30 individuals analyzed their reliability by determining their confidence in the innovation's behavior evaluation form and satisfaction assessment form. Cronbach's alpha coefficients of 0.87 and 0.89, respectively, and the observation form for waste management from bedridden patients in the community was used for qualitative analysis. Frequency, percentage, mean, and standard deviation were used for the descriptive statistics.

Human research ethics The Chaiyaphum Provincial Public Health Office Research Project Number EC 65/2020 was approved by the Human Research Ethics Committee. The duration of Phases 1 through 4 of the research was one year and seven months (1 November 2020–31 March 2022).

The numerous procedures and activities that were performed are detailed in Figure 2, along with the samples utilized at each stage of the process.

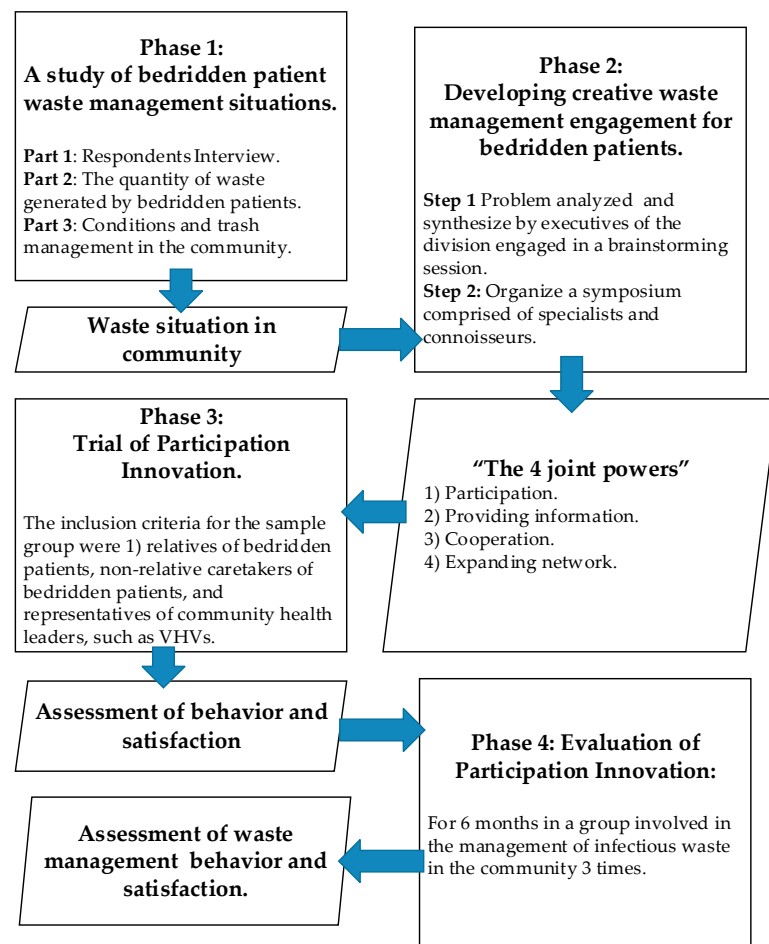

**Figure 2.** Provides a summary of the research investigations conducted throughout each step of the study as well as the research findings.

### 4. Results

The findings of Phase 1 research on the situation and waste management of bedridden people.

Based on interviews with 78 caregivers of bedridden patients, the total waste from bedridden patients in the community in areas was 893 kg/week, according to the findings of a survey, separated into adult diapers (546 kg/week), urine catheters (6 kg/week), toilet paper (85 kg/week), cotton (61 kg/week), bandage (78 kg/week), dialysis fluid lining pad bag (15 kg/week), gloves (8 g/week), cotton swabs (46 kg/week), food hose (6 kg/week), syringes (3 kg/week), and suction hose (4 kg/week) are required for the handling of human waste, sick in bed, it was discovered that all waste was gathered in a red bag and taken to the health-promoting center or burned in the home. Based on interviews with 110 people who were close to bedridden patients, the following waste management problems and challenges were identified:

Bedridden patients generate more waste in the community, and there is no separation of bedridden waste from normal waste. This may have been the result of uninformed caregivers regarding waste segregation. In addition, disposal facilities are inadequate. The waste of bedridden patients is combined with general waste because it must be disposed exclusively within the hospital. In addition, the community lacks an understanding of the hazards and impacts of waste from bedridden patients, community participation is restricted, and there is no defined procedure for handling waste from bedridden patients.

During brainstorming sessions, guidelines for managing waste from immobile patients were formulated. Waste management rules for immobile patients have been developed to address community concerns and needs. The community's problems and needs can be summarized by the term "responsibility", which refers to the process of establishing whether an individual is accountable for his or her own acts. Therefore, it has been suggested that gloves, red bags, and masks should be purchased to collect excrement from immobile patients. Due to a lack of educated staff, caregivers and family members of bedridden patients were required to complete training on how to sort solid waste into categories such as medical waste, general waste materials, and disease-spreading waste.

1. It was less involved in waste management; thus, it was suggested that a project should be conducted to improve awareness of domestic waste management in the community and waste management for bedridden patients, including household waste segregation and clean-house competition. By encouraging communities to efficiently separate waste, the system was able to provide a more secure environment for waste storage.
2. Waste management knowledge of family members, caregivers, and VHVs of bedridden patients: they were uninformed about the types of waste, how to sort them, collect them (collect them in the same bags as regular home waste), and how they were delivered. Therefore, the community must be educated on how to distinguish between typical waste that might transmit illness and how to properly dispose of it.

*The Issue of the Location of Waste Disposal for Immobile Patients*

Due to the remote location of sub-district health promotion hospitals, it is difficult to dispose of the infectious waste created daily by bedridden patients, resulting in an increase of infectious waste in the homes of bedridden patients. Therefore, infectious waste should be removed straight from the patient's residence and disposed of in a vehicle.

Phase 2: Developing creative waste management engagement for bedridden patients.

Forty stakeholders in bedridden patient waste were given a two-stage plan on the problem and necessity for waste management. The forum recognized local administrative organizations and two brainstorming forums for people directly involved in the waste management of bedridden patients in the community, to comment and return information on the waste management of bedridden patients in the current community, and to develop a strategy for participatory waste management of bedridden patients. Evaluating and synthesizing online forum group discussions. The four power joints introduce those with limited waste management movement. Waste requires community engagement to examine problems and waste management; creating community knowledge, attitudes, and competence in waste management for each type of bedridden patient; engaging in critical review; and benefiting from recommendations. Determine the day and time for collecting and delivering infectious waste from bedridden patients. Recognize community agreements, including bedridden patients' waste management. Four Joints of Power enhanced the waste management network by increasing the network of adjacent villages and holding training to build a unique strategy for bedridden patients to participate in waste management based on their professional experience. The researcher was chosen from among 15 waste management professionals to provide additional advice. Mutual acceptance and community coverage may render this paradigm more inclusive and implementable. Summary of meeting outcomes: Most participants made recommendations on how to apply the invention and its benefits to improve waste management among bedridden patients.

Most respondents recommended implementing the idea based on its popularity. These innovative components are described below. Most of those who covered innovation thought that it would improve the system's performance. They have defined purposes and meet the community's intended demographics, thus allowing for broader growth.

Phase 3: Trial of Participation Innovation.

The researcher has executed Phase 2 innovations in the community of Khok Mang Ngoi sub-district, Khon Sawan district, which has the highest number of bedridden patients,

and all parties concerned have voluntarily tried the innovation. The four power joints consist of four steps.

(1) Participation (learning how to manage waste): This includes actions to improve awareness, such as recognizing waste problems, managing bedridden patients' waste, raising community knowledge and attitudes, and educating about waste management, waste creation, and its effects. Workshops improve waste management participation, knowledge, and positive attitudes. The project participants were divided into four groups of ten for the discussion. An organization that identifies immobile people's issues, causes, and remedies to empower the community. The results showed the origins and impacts of bedridden waste.

(2) Analyze concepts, practices, and agreements on waste management produced by bedridden patients in the community depending on the type of waste, including designating a day and time for infectious waste collection and disposal. The community's shared responsibility for waste management allowed them to manage waste problems, reduce the impact of waste from immobile patients, and promote a new culture of innovation in the Khok Mang Ngoi sub-district.

(3) Information and implementation of community agreements according to mutually agreed-upon guidelines, including the following activities: training in the separation of household and patient waste, management of hazardous and infectious waste separation systems from general waste at the source, collection of infectious waste from bedridden patients to be handed over to create fertilizer from organic waste, and participation in the operation and appreciation of the community's agreement by collectively managing bedridden patient waste.

(3.1) Residential and bedridden patients' medical waste is segregated, and waste sorting enables source separation. They separated the infected waste from normal waste and collected it from immobile patients.

(3.2) Organic waste was isolated from other wastes and saved by developing knowledge and skills regarding separation and use of organic waste to make biofertilizers at home. Agriculture can also benefit from biofertilizers.

(4) Increasing waste management networks for bedridden patients in surrounding regions by encouraging bedridden patients' caregivers or relatives to join infectious waste management activities for bedridden patients in the community.

The community was able to handle waste from bedridden patients in agreement with the 4 Joints of Power as a consequence of the development of innovation. Table 1 displays a statistically significant difference at the 0.05 level between the average waste management knowledge of bedridden patients, which was 10.35 points prior to development, and the average knowledge of 13.23 points after development.

**Table 1.** A comparison of the waste management knowledge of relatives, non-relatives, and community health volunteers before and after the implementation of the 4 Joints of Power innovation ($n = 40$).

| Waste Management Knowledge | $\overline{X}$ | SD | df | T-Value | *p*-Value |
|---|---|---|---|---|---|
| Before knowledge | 10.35 | 1.26 | 39 | −16.32 | $p < 0.05$ * |
| After knowledge | 13.23 | 1.12 | | | |

Note: * There was a difference at the 0.05 significance level.

Before development, the average attitude toward waste management among the bedridden patients in the target group was at the 0.05 level; there was a statistically significant difference between the mean attitude toward waste management before and after the development, which was 3.75 and 4.14, respectively (as shown in Table 2).

**Table 2.** Compares the attitudes of VHVs, family members, and non-relatives toward waste management before and after using the four Joints of Power innovation ($n$ = 40).

| Waste Management Attitudes | $\overline{X}$ | SD | df | T-Value | $p$-Value |
|---|---|---|---|---|---|
| Before attitudes | 3.75 | 0.85 | 39 | −4.87 | $p < 0.05$ * |
| After attitudes | 4.14 | 0.42 | | | |

Note: * There was a difference at the 0.05 significance level.

Phase 4: Evaluation of Participation Innovation.

In Phase 3, bedridden families, bedridden caregivers, and diverse community leader representatives reviewed the 4 Joints of Power. Six months later, the study team assessed infection and waste management by implementing the innovation and evaluating waste separation behavior and sample satisfaction following three applications of the innovation. The findings of this study are presented in Tables 3 and 4.

**Table 3.** Mean, SD, and Interpretation of separation, storage, and disposal of medical waste ($n$ = 60).

| Contagious Waste Management Conduct | $\overline{X}$ | SD | Interpretation |
|---|---|---|---|
| Separation and storage of medical waste | 4.25 | 0.51 | Excellent |
| Disposal of infectious waste | 4.27 | 0.42 | Excellent |
| Total | 4.26 | 0.47 | Excellent |

**Table 4.** Number and percentage of overall satisfaction with waste management approach bedridden patients ($n$ = 60).

| Satisfaction | $n$ | % |
|---|---|---|
| Extremely satisfied level | 56 | 93.50 |
| Moderate satisfied level | 4 | 6.50 |
| Total | 60 | 100.00 |

An evaluation of those participating in waste management determined, as indicated in Table 3, that the waste management behavior of sixty bedridden patients was excellent, and there were progressively significant ($\overline{X}$ = 4.25) and ($\overline{X}$ = 4.27) developments. Overall, the participation of bedridden patients in the innovation of waste management is considerable. 4.26 on average.

As Table 4 displays the number and percentage of bedridden patients, as well as their overall satisfaction with the medical waste management model. It was determined that 93.50 % of patients had the highest satisfaction, followed by 6.50 % with moderate satisfaction.

## 5. Discussion

The results of the study were as follows: participation of immobile patients in innovative waste management (4 Joints of Power) reduced the community's infectious waste problem from immobile patients. The procedure consisted of four phases: Phase 1, a study of bedridden patient waste management situations; Phase 2, two stages for using waste as an invention with family members, non-relative caregivers, VHVs, and community leaders: (1) analyzing the problem or its cause and generating management alternatives through collaborative brainstorming with community members; and (2) gathering the thoughts and suggestions of a number of agency specialists. An outcome is a novel approach to waste management by bedridden patients known as "Four Joins of Power", which includes (1) involvement in events and boosting community knowledge and attitudes, and (2) providing information on each type of waste. (3) cooperation in waste management (analysis, planning, and execution) and abiding by established norms, and (4) expanding the waste

management network. Phase 3 was the participation innovation trial, and Phase 4 was the evaluation. Based on the findings of this study, it was concluded that innovation affected the community's behavior and level of collaborative efforts. Community waste separation techniques have evolved since the beginning, and the waste produced by bedridden patients has been managed. The Findings of previous studies show that the innovative success factor depends on.

1. Social support was made up of leading factors such as knowledge, attitudes, and behaviors about reducing infectious waste; contributing factors such as having a local solid waste management policy; and supplementary factors such as obtaining information about reducing solid waste from public health officials and VHVs in different ways, such as getting help in the form of information, materials, or emotional support. The use of innovation has generated a new waste management approach that can be implemented in the community to aid waste reduction. The results of this study showed that these changes affect how people in a community act and how much they work together. Since the beginning, waste sorting procedures in communities have changed and domestic waste from bedridden patients has been under control. Ban Prasat Amnuay Muang Fai sub-district, Nong Hong district, Buriram province discovered that family leaders had higher average scores on waste management knowledge, awareness, and involvement after participating in the process than before participation. Significant at the 0.001 level, each household was mandated to separate waste and dispose of it according to the management principles. This includes refraining from dumping waste into the public spaces of the village. Community regulations on reuse waste management exist. Comparing the quantity of medical waste before and after adopting the waste management model with community involvement, the study found that the amount of waste after applying the model was statistically significant at the 0.05 level. Therefore, participation from all community sectors is essential for the formulation of an efficient waste management plan for the community. Owing to mismanagement and lack of treatment, healthcare waste management (HWM) in developing nations frequently poses a threat to human health and the environment [1,36,37]. The absence of effective activities for healthcare waste (HW) minimization, separation, and recycling [38,39], low levels of training and awareness of waste legislation [40], increased the spread of diseases [41], and decreased the quality of the service provided and the safety of the operators [38,42]. Determining the generation rates of HW in hospitals requires consideration of the number of beds [43]. The increase in the use of disposable medical items and the growth of the global population contribute to the rise in HW generation [39], which exacerbates waste management challenges in low- and middle-income nations. The range of HW production in low-middle-income nations might range from 0.02 kg bed/day to 3.2 kg bed/day, depending on the vast differences between rural, suburban, and urban healthcare systems, the number of occupied beds, and the country's revenue [44]. The main factors are the lack of data on waste generation and the absence of programs for waste minimization, suitable treatment, and educated employees, which negatively impact HWM planning [37,45,46].

2. Local policy on waste management: In Thailand, the Ministry of the Interior is responsible for determining the overall waste management policy at the operational level; therefore, local government organizations should prioritize allowing local authorities to participate in operations and decentralize management authority. As part of an integrated management and networking approach (governed by a network), the provincial governor and the sheriff must take a leadership role in formulating community waste management plans and initiatives as well as fostering collaboration and coordination with government agencies and the public and private sectors (joined-up government). In addition, the waste management policy in the community must have plans or projects to raise the awareness of the people and local government executives involved in various forms of waste and to promote public participation and public responsibility, such as allowing people in the community to participate in the project and to have

a common agreement, such as a village statute, as well as a policy to use modern technology and digital platforms, such as a village homepage. To reduce the quantity of infectious waste in the community and among families, relatives, and caregivers of bedridden patients, it is essential to have a proper communication channel and inquire with health workers and VHV about the amount of infectious waste in the community [45].

3. Social engagement: This is a learning process for all departments and communities to "explode from the inside" together to learn and create an understanding of the causes of problems and ways to solve them by themselves, including the measurable social impact that can be assessed (scholarship). Measurement of the impact of changes that occur in economic, social, and policy aspects that are clearly measurable to change for the better. This study determined that staff participation in the development of educational packages and the delivery of educational classes contributed to the successful implementation of the interventions. Separation and collection of medical waste are the most crucial phases of waste management. Previous research in Iran has demonstrated that only 25% of primary healthcare facilities adequately segregate hazardous waste [47]. In community health centers (CHCs), medical waste must be separated at the point of generation [47,48]. Therefore, it must provide the required facilities and equipment for standard separation, such as color-coded bags, containers, and bin labeling. Engaging activities should be developed to equip hospital workers with information on the subject. Aroonsri and Phatisena (2021) [49] conducted a study to encourage the participation of health partners and households. Before participating in the demonstration, it was discovered that most participants did not segregate their waste. Instead, they have utilized waste disposal technologies involving the combination and incineration of waste. After participating in the learning process, home leaders had statistically higher scores for knowledge, awareness, and participation in waste management. People segregated their waste and disposed of it correctly according to all household management principles, including refraining from littering in the village's public areas. They have created waste management regulations for the community's future use. Community members contributed exceptionally well to every aspect of waste management. The effect of involvement was the engagement in activities. The organization has designed its community leaders to prioritize the participation of individuals in every aspect, beginning with the provision of information, analyzing problems, providing obstacles and requirements, and choosing viable solutions. Alternatively, community leaders must support and promote the requirements of organizational activities. Involvement in waste management was encouraged at each level of inquiry.

4. Sharing and learning: Learning exchanges enlighten, inspire, and generate new experiences and knowledge to assist participants in learning, knowing, and comprehending in a methodical manner. In one location, there was a behavior transformation, particularly in managing infectious waste in the community, as well as developing skills and knowledge. It also produces new information, inventions, and expansion of ideas, all of which contribute to the success of an activity. The findings of a study that was carried out in Iran, one of the most significant issues that plague healthcare systems is a shortage of staff members who are knowledgeable about waste management. They concluded that the initial compliance rates for management and training were 22.8% and 41.0 %, respectively. A significant factor is creating a body of knowledge for caregivers of bedridden patients regarding the separation of infectious waste by sorting, collecting, and referring to appropriate authorities. This was consistent with the findings of Amouei et al. (2015) [50] which indicated that hospital workers possessed a poor degree of knowledge. Literature indicates that individuals with inadequate knowledge cannot operate effectively. According to Rujirat (2019) [51], there is a lack of precise attitude and comprehension, awareness, and continual public relations regarding infectious waste management to develop knowledge. Waste

disposal information should be delivered in all forms of multimedia. The main goal is to provide handouts and printed materials, such as environmental publications, particularly those on infectious waste and healthcare, to communities. According to evidence, the most significant factors contributing to the failure of temporary storage in Iran are the absence of a physical site, noncompliance with the standard duration of infectious waste in CHCs, and the low level of education of waste workers responsible for the collection and transportation of medical waste [46].

Based on the aforementioned study, the research team described the relevant variables that will promote sustainability for the community in lowering infectious waste in the community as follows:

1. Community engagement depends on the neighborhood's growth potential. The community is a key objective for many activities when it can create power and be self-sufficient. Knowledge and experience analyze challenges and suggest answers based on the concept that everyone is a viewpoint count, a fundamental process that provides participants with a sense of belonging at each level and enables them to take action based on community demands and realities.

2. The local administrative organization should establish and implement the following strategies for managing infectious waste in the community:

   2.1. Upstream measures, such as public relations campaigns, encourage people to separate infectious waste in the community, such as face masks, cloth masks, and waste contaminated with mucus, saliva, or secretions, to separate waste bags from other types of waste, close their mouths, and use the symbols of the waste bin.

   2.2. Midway measures, local government agencies should provide red bins in public areas or at waste collection sites as designated locations for infectious waste in the community, as well as anti-infection equipment such as gloves, face masks, cloth masks, and protective clothing. This is concise for the uniform operation of infectious waste collection workers.

   2.3. Destination measures, such as the use of laws to dispose of infectious waste, such as the Ministerial Regulation on Disposing of Infectious Waste, B.E.

## 6. Conclusions and Future Research

Phase 1: The amount of waste in the community increased due to a lack of waste segregation, lack of recycling, and insufficient containers for waste disposal, according to the findings of a Phase 1 study of the situation and waste management of bedridden patients in the community based on interviews with those involved. Community members do not have a thorough understanding of the hazards and implications of excessive waste and waste management systems, nor do they influence them.

Phase 2: Developing creative waste management engagement for bedridden patients. The researcher presented stakeholders in bedridden waste management with a recommendation on the problem condition and need for waste management among the community's bedridden patients. By organizing a platform for executive brainstorming representatives of local groups and local administrative organization representatives, the result was a draft innovation for involvement in waste management from bedridden patients, titled "4 Joints of Power", which is: (1) participatory activities and enhancing community knowledge and attitudes, and (2) providing information on the management of each type of waste. (3) Cooperate in waste management (analytical thinking, planning, execution, etc.) and are regulated by mutually agreed-upon rules. (4) Expanding the waste management network.

Phase 3: Trial of Innovation.

The Phase 2 study-derived invention "4 Joints of Power" was implemented in accordance with the following innovation guidelines: Caregivers of bedridden patients in the demonstration community possessed knowledge and attitudes regarding waste separation and organic waste management in various forms, such as preparing organic fertilizer, determining the date and time for collecting infectious waste and sending it for disposal,

and constructing a network to manage waste from bedridden patients. The community's knowledge ($\overline{X}$ = 13.23 levels) and attitudes ($\overline{X}$ = 4.14 levels) on waste management among participants significantly increased at 0.05 levels.

Phase 4: Following the implementation of the "4 Joints of Power" innovation, they did their waste management behavior of collecting ($\overline{X}$ = 4.25) and delivering waste to disposal ($\overline{X}$ = 4.27), and the majority of those who participated in waste management were satisfied (93.50%).

Solving the problem of increasing medical waste for bedridden patients in major communities requires the participation of the community residents in the analysis to obtain evidence-based data and the management of strategies by leaders and government officials regarding the situation of medical waste to increase the community's awareness and comprehension of the consequences of the increasing amount of medical waste. The waste management agreement and expansion of the medical waste management network in the community focus on the correct separation of waste before it is thrown away as well as people's practices regarding the proper separation of infectious waste, which will support the sustainable management of medical waste in the community.

The "4 Joints of Power" innovation consists of 4 parts: (1) community engagement; (2) providing knowledge on how to handle infectious waste in the community; and (3) community, workplace, and individual collaboration. (4) The network strength of social movements, academic institutions, and communities. Phase 3 is the outcome of investigations that have been proven and developed in line with Phases 1 and 2. The amount of infectious waste produced by bedridden patients serves as the starting point for data analysis, for stakeholders to assess and address problems, and to acquire knowledge of infectious waste management. Phase 3 was effective in assessing stakeholder knowledge and attitudes. In Phase 4, waste classification behavior was evaluated to improve the precision of waste classification. Thus, the 4 Joints of Power can be considered innovations in the process of infectious waste management for immobile patients.

### 6.1. Suggestions for Using the Study's Findings

(1) Government organizations that collect municipal waste or local government agencies that are part of this study can use the novel method of bedridden patients participating in waste management to support bedridden patients in waste management and promote waste management in other communities.

(2) Local government units must have a waste management system. In addition to disposing of waste or reusing waste based on the type of waste encountered in the community context within the correct management of waste according to academic standards, waste collection systems have been developed to reduce the amount of waste disposal and the impact of community waste problems on quality of life and the environment.

(3) Municipal government organizations set up rules or ordinances for managing waste in a way that is good for the environment based on input from the community.

### 6.2. Suggestions for Future Investigation

It is crucial to compare the four Joints of Power, a revolutionary approach to bedridden waste management, with existing methods that consider social, psychological, economic, and demographic issues. Social impact evaluation and caregiver awards for innovative waste management from immobile patients will assist communities in overcoming infectious waste management problems.

**Author Contributions:** S.P.: Original draft preparation, Conceptualization, Methodology, Software; S.C.: Supervision, Data Curation and Writing Validation; C.N.T. wrote, reviewed, and edited the manuscript. All authors have read and agreed to the published version of the manuscript.

**Funding:** This research was funded by the Health, Environment, and Clean Energy Promotion Institute Foundation and Office of the National Health Commission.

**Institutional Review Board Statement:** Not applicable.

**Informed Consent Statement:** Not applicable.

**Data Availability Statement:** Not applicable.

**Acknowledgments:** This study was part of a project titled "The Innovation of Community Participation in Waste Management of Bed-Bound Patients in Chaiyaphum Province, Thailand". We thank the organization leaders, VHVs, and relatives of the bedridden patients who attended the conference and provided vital information. The authors are grateful to the Health, Environment, and Clean Energy Promotion Institute Foundation and Office of the National Health Commission, Thailand.

**Conflicts of Interest:** The authors declare no conflict of interest.

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
