# Peer review of "“Four Joints of Power” Innovation of Community Involvement in Medical Waste Management of Bed-Bound Patients in Thailand"

_sustainability, doi:10.3390/su15021669_

Round 1
Reviewer 1 Report (Previous Reviewer 3)
The author has addressed all the comments, and this paper can be considered at the current form.
Author Response
Dear Reviewer
Thank you for your kindness in providing suggestions and improvements in various parts to make the manuscript better quality, which the research team has now revised according to the suggestions as the corrections and improvements of the research team are written in red letters.
Best regard
Dr. Chaiwong
Reviewer 2 Report (New Reviewer)
Dear Authors,
in my opinion, the paper is well written. The topic is interesting and the text is very fluent . In the introduction I suggest to highlight more the added value of the research. It is mentioned on the third page, but it would be appropriate to see if there are similar studies (although in the final part the study in Iran is reported). On line 72 of the second page there is a typo. I think the year refers to 2020 and not 2030! The methodology is clear although it would also be interesting to indicate the time reference in which the study was carried out. The results are effectively described. The discussions are clear. The conclusions appear well argued.
Author Response
Dear Reviewer
I would want to express my gratitude to the reviewer for providing me with helpful feedback on how to improve the quality of my writing and for spreading the word about the advantages of reading this article. For your recommendation, I respond it the fixes and improvements made will be written in black letters and highlighted in gray and the corrections and improvements of the research team write in red letters
Reviewer 2
in my opinion, the paper is well written. The topic is interesting and the text is very fluent. In the introduction, I suggest highlighting the added value of the research. It is mentioned on the third page, but it would be appropriate to see if there are similar studies (although in the final part the study in Iran is reported). On line 72 of the second page, there is a typo. I think the year refers to 2020 and not 2030! The methodology is clear although it would also be interesting to indicate the time reference in which the study was carried out. The results are effectively described. The discussions are clear. The conclusions appear well-argued.
Answer: I changed it from “In 2030, there were 152,749 bedridden groups.” to “there will be 152,749 bedridden groups” in line 74. Because it is predicted bedridden groups in 2030 in lines 75-76.
Best regard
Dr. Chaiwong
Reviewer 3 Report (New Reviewer)
· The entire abstract must be revised and particularly, in the Line 18 of the abstract section, a full stop must be deleted, and the entire sentence rephrased.
· The Line 24 of the abstract section should also be rephrased.
· The entire Introduction must be revised as Lines 61-63 and Line 91-92 are repetitions.
· Manuscript lacks novelty, please state the novelty of your study, and please shorten the Introduction by deleting all the repeated statements as they are not adding values to the manuscript.
· The service of a language expert must be sought as many sentences were incoherent.
Overall, this manuscript is significant, however, at this stage I do not recommend its publication until the entire manuscript is revised and all issues raised are addressed by the authors.
Author Response
- Dear Reviewer
I would want to express my gratitude to the reviewer for providing me with helpful feedback on how to improve the quality of my writing and for spreading the word about the advantages of reading this article. For your recommendation, I respond to it with the fixes/improvements writing black text with yellow highlights.
Your comment
The entire abstract must be revised and particularly, in the Line 18 of the abstract section, a full stop must be deleted, and the entire sentence rephrased.
Answer: I rewrite a new abstract
The purpose of this study was to encourage innovative participation in the management of medical waste by bedridden patients in the research region of Khon Sawan, Chaiaphum Province, through research and development. The steps are as follows: Phase 1: A study of bedridden patient waste management situations. using the amount of waste generated as an innovation with relatives, non-relatives, village health volunteers (VHVs), and community leaders. Phase 2: Developing creative waste management engagement requires two steps: 1) analyzing the problem or its cause and generating management alternatives through collaborative brainstorming with a community member, and 2) gathering the thoughts and suggestions of a number of agency specialists. The outcome is a novel model of participation in waste management by bedridden patients termed "4 Joins of Power," which includes 1) participatory activities and enhancing community knowledge and attitudes, and 2) providing information on the management of each type of waste. 3) cooperation in waste management (analytical thinking, planning, execution, etc.) and are regulated by mutually agreed-upon rules. 4) expanding the waste management network jointly. Phase 3 is the innovation trial, while Phase 4 is the innovation assessment. Using the paired t-test to compare pre-and post-development knowledge and attitudes, and conducting qualitative data analysis. In Phase 3, after implementing the collaborative innovations, the average knowledge ( =13.23) and attitudes( =4.14) regarding waste management increased considerably (P<.05), and in Phase 4, waste management behavior comprising sorting, storage, and disposal was observed. There were progressively substantial gains ( =4.25 and =4.27, respectively). 93.50% of the most collaborative individuals were satisfied. In order to reduce the amount of waste that must be sorted and collected, it is necessary to emphasize from the commencement the participation of people and networks from all sectors in the area in joint thinking, planning, and comprehensive analysis ensure the sustainability of waste management in the community.
- The Line 24 of the abstract section should also be rephrased.
Answer: I rewrite a new abstract
- The entire Introduction must be revised as Lines 61-63 and Line 91-92 are repetitions.
Waste management is a major concern in the majority of communities, particularly in developing nations lacking environmental education initiatives in line 61-63.
Waste disposal is a serious challenge in the majority of communities, particularly in developing nations without environmental education initiatives [12] in line 91-92.
I rewrite a new sentence
“In the majority of communities, waste management is a serious challenge, particularly in developing nations without environmental education campaigns” [11,14-15] in line 63-65.
- Manuscript lacks novelty, please state the novelty of your study, and please shorten the Introduction by deleting all the repeated statements as they are not adding values to the manuscript.
The "4 Joints of Power" innovation consists of 4 parts: 1) community engagement; 2) providing knowledge on how to handle infectious waste in the community; and 3) community, workplace, and individual collaboration. 4) the network strength of social movements, academic institutions, and communities. Phase 3 is the outcome of investigations that have been proven and developed in line with Phases 1 and 2. The amount of infectious waste produced by bedridden patients serves as the starting point for data analysis and for stakeholders to assess and address problems, as well as acquire knowledge of infectious waste management. Phase 3 was effective in assessing the knowledge and attitudes of stakeholders. In phase 4, which was to improve the precision of waste classification, the behavior of waste classification was evaluated. The 4 Joints of Power can thus be considered an innovation in the process of infectious waste management for immobile patients in line 749-759.
Overall, this manuscript is significant, however, at this stage I do not recommend its publication until the entire manuscript is revised and all issues raised are addressed by the authors.
Best Regard
Dr. Chaiwong
Reviewer 4 Report (New Reviewer)
The paper presents engagement with bedridden communities in Thailand in an effort to raise their awareness of waste management. As a community activity the work has good merit, but as a scientific effort this paper has many problems:
1. This does not seem to be a scientific or academic paper. The overall structure of the paper is not formatted as a scientific manuscript and the content does not appear to be academic.
2. The methods section does not sufficiently describe the research methods applied. There is no information about the questions posed to the selected members. There is a repetition about the data analysis. The data analysis seems to be weak.
3. The abstract does not clearly summarise what was done and what are the major outcomes.
4. There are many grammatical and language mistakes and improper sentences.
5. How were the quantitative and qualitative data collected in line 133?
6. What does "participation in familiar and community" mean in line 145?
7. Why was the data analysed after 6 months in line 248?
8. How different is the "Data analysis" in lines 249-251 and lines 260-263?
9. How was 893 kg/week waste in line 278 determined? Is this accross 78 caretakers?
10. Lines 305 and 312 have incomplete sentences
11. What is meant by "return of information to obtain information" in line 332?
12. What is meant by "The dog searchers through the trash" (lines 405-406), where is this observation coming from, how it relates to the text before and after?
13. How was the p-value in Tables 1 and 2 determined and how is it 0? Was this performed for the median results or all results? It doesn't seem 3.75 and 4.14 values in Table 2 are significantly different and they appear within the margin of deviation. It doesn't seem the statistical analysis was correctly performed.
Author Response
Dear Reviewer
Thank you for your kindness in providing suggestions and improvements in various parts to make the manuscript better quality, which the research team has now revised according to the suggestions as Fixes/improvements written in black text with yellow highlights, and the corrections and improvements of the research team are written in red letters.
Best regard
Dr. Chaiwong
Your Suggestion
The paper presents engagement with bedridden communities in Thailand in an effort to raise their awareness of waste management. As a community activity the work has good merit, but as a scientific effort this paper has many problems:
- This does not seem to be a scientific or academic paper. The overall structure of the paper is not formatted as a scientific manuscript and the content does not appear to be academic.
Answer: I was in charge of organizing the paper formation and following the editorial board's advice.
Following the formatted as a scientific manuscript by the editorial board as
- Introduction
- Literature Review
- Materials and Analysis
- Result
- Discussion
- Conclusions and Future Research
- The methods section does not sufficiently describe the research methods applied. There is no information about the questions posed to the selected members. There is a repetition about the data analysis. The data analysis seems to be weak.
Answer: I revised the methodology of the study as well as the classifications of the materials and the analysis (data collecting technique, population/sample, data collection instrument, and statistics), and I increased the lengths of the study for each phase from lines 242–384.
- The abstract does not clearly summarise what was done and what are the major outcomes.
I added a clearly summarise of MS with the process of the 4 Joints of Power” as
“The outcome is a novel model of participation in waste management by bedridden patients termed "4 Joins of Power," which includes 1) participatory activities and enhancing community knowledge and attitudes, and 2) providing information on the management of each type of waste. 3) cooperate in waste management (analytical thinking, planning, execution, etc.) and are regulated by mutually agreed-upon rules. 4) expand the waste management network jointly” in lines 17-23.
- There are many grammatical and language mistakes and improper sentences.
I checked all manuscripts.
- How were the quantitative and qualitative data collected in line 133?
I added “Quantitative data included general data and the amount of waste generated by bedridden patients, while qualitative data included the circumstances of bedridden patients and waste management in the community” in lines 260-262., and showed the research instrument as
The research instrument was a three-part semi-structured interview.
Part 1: General data
Part 2: the quantity data of waste generated by bedridden patients (in one week)
Part 3: Bedridden patients' conditions and waste management in the community. There were five inquiries:
1) the current situation of the waste problem caused by immobile people and an assessment of the community's future condition.
2) the criteria for the creation of a community waste management system for bedridden patients.
3) participation in familiar and community
4) the knowledge, understanding, and awareness of waste management among bedridden patients in the community.
5) the behavior of waste management among bedridden patients at home. Each individual is interviewed for no longer than forty minutes in lines 273-286.
- What does " participation in familiar and community " mean in line 145?
I changed from “participation in familiar and community” to “Family members of bedridden patients, non-relative caregivers, and VHVs in lines 278.
- Why was the data analyzed after 6 months in line 248?
Answer: I followed the theory of the stages of change as
“Phase Four: Taking Action/Mustering Your Willpower
People are at this stage when they are motivated to alter their behavior and are actively involved in taking measures to modify their poor behavior by employing a number of various strategies. This is the stage where change is most likely to occur. This is the quickest and easiest part of the whole process. There is no standard length of time that individuals spend engaged in an activity. In most cases, it lasts for roughly six months, but in certain cases, it might be as brief as one hour! When people are at this stage, they are most reliant on their own strength of will. They are trying very hard to modify the habit, but they are also the ones that have the biggest chance of failing.
- How different is the "Data analysis" in lines 249-251 and lines 260-263?
Answer: I rewrite to new sentence “Frequency, percentage, mean, and standard deviation were utilized to explore descriptive statistics, and a t-test was performed to analyze persons' knowledge and attitude toward medical waste management before and after significant improvement. 05 levels.
- How was 893 kg/week of waste in line 278 determined? Is this across 78 caretakers?
Answer: I interviewed 78 caretakers and collected the waste produced by bedridden in their households.
- Lines 305 and 312 have incomplete sentences
The old sentence was
“During brainstorming sessions, guidelines for managing waste from immobile patients were formulated. The concerns and requirements of the community can be summed up as follows:
Problems with waste disposal from bedridden patients
1) The community lacks the materials and equipment required to properly handle medical waste. Therefore, it has been advised that gloves, red bags, and masks be purchased so that the waste of immobile patients can be collected.
2) Due to a lack of trained workers, caretakers and family members of bedridden patients were compelled to undergo a course on how to sort solid waste into categories such as medical waste, general waste, and disease-spreading waste.
Problems with bedridden individuals' waste management, including the community
Fewer community members participate in waste management. Therefore, it was proposed to establish a community-wide project to increase awareness of household waste management from bedridden patients, including a residential waste sorting competition, establish a trash-free community project throughout the community, such as a participatory waste management project, zero-waste model home construction, etc.”
I rewrite it to “During brainstorming sessions, guidelines for managing waste from immobile patients were formulated. Waste management rules for immobile patients were developed to address the community's concerns and needs. The community's problems and needs can be summarized by the term "responsibility," which refers to the process of establishing whether an individual is accountable for his or her own acts. Therefore, it has been suggested that gloves, red bags, and masks be purchased in order to collect excrement from immobile patients. Due to a lack of educated staff, caretakers and family members of bedridden patients were required to complete training on how to sort solid trash into categories such as medical waste, general waste materials, and disease-spreading waste in lines 406-414.
.
- What is meant by "return of information to obtain information" in line 332?
I changed the sentence from “The forum recognized representatives of local administrative organizations, as well as 2 brainstorming forums for groups of people who are directly involved in the waste management of bedridden patients in the community, to comment and return information to obtain information about the problem and waste management of bedridden patients in the current community and to develop a strategy for participatory waste management of bedridden patients” to “The forum recognized representatives of local administrative organizations, as well as 2 brainstorming forums for a variety of people directly involved in the waste management of bedridden patients in the community, to comment and return information on waste management of bedridden patients in the current community and to develop a strategy for participatory waste management of bedridden patients” in line 434-438.
- What is meant by "The dog searchers through the trash" (lines 405-406), where is this observation coming from, how it relates to the text before and after?
I rewrite a new sentence in line 471-509.
- How was the p-value in Tables 1 and 2 determined and how is it 0? Was this performed for the median results or all results? It doesn't seem 3.75 and 4.14 values in Table 2 are significantly different and they appear within the margin of deviation. It doesn't seem the statistical analysis was correctly performed.
Answer; In Tables 1 and 2, calculations compare the means of knowledge (Table 1) and attitude (Table 2) of using the Four Joints of Power for innovation using statistics. The t-test was calculated by using an automatic statistical program that made the p-value obtained in both tables 0.00. As a result, in Tables 1 and 2, the p-value was changed to.05 rather than.00. According to the comparison results, the mean before and after knowledge (Table 1) and attitude (Table 2) of using the Four Joints of Power innovation were significantly different at level.05 (p<.05).
Reviewer 5 Report (New Reviewer)
The study reported the innovative engagement in the management of medical waste by bedridden patients in the research district of Khon Sawan, Chaiaphum Province, using a research and development approach. The manuscript was well prepared. Some suggestions are shown below to improve the quality of manuscript.
In Abstract, how the findings from this work guide the medical waste management of bed-bound patients in Thailand should be briefly described at the end of Abstract.
Section “2. Population” and section “3. Step of the research process” should be combined into one section such as “project design”. Some description such as “Data analysis” should be combined. Moreover, figure 1 should be described in “project design” rather than section “Introduction”.
“4. The result” should be revised “4. Results”.
In the manuscript, the description “3.1 Depending ……”, “3.2 Making home ……” should be revised because it easily misleads readers as sub-section of section “3.”.
There is less information in Table 1- table 4, some tables should be combined or should be described in words rather than in table.
At the end of section “5. Discussion”, brief discussion of how the findings from this work guide the medical waste management of bed-bound patients in Thailand should be added.
Some references are suggested to support the description of waste and pollutant removal: A review of disposable facemasks during the COVID-19 pandemic: A focus on microplastics release, Chemosphere, 2022, 137178; MXenes as Heterogeneous Fenton-like Catalysts for Removal of Organic Pollutants: A Review, Journal of Environmental Chemical Engineering, 2022, 108954; Metal-organic frameworks-derived catalysts for contaminant degradation in persulfate-based advanced oxidation processes, Journal of Cleaner Production, 2022, 134118; A review on persulfates activation by functional biochar for organic contaminants removal: Synthesis, characterizations, radical determination, and mechanism, Journal of Environmental Chemical Engineering 2021, 9 (5), 106267.
The Conclusions section is redundant and should be refined.
Author Response
Dear Reviewer
Thank you for your kindness in providing suggestions and improvements in various parts to make the manuscript better quality, which the research team has now revised according to the suggestions as Fixes/improvements written in black text with yellow highlights, and the corrections and improvements of the research team are written in red letters and the corrections and improvements of the research team are written in red letters
Best regard
Dr. Chaiwong
Your Suggestion
The study reported the innovative engagement in the management of medical waste by bedridden patients in the research district of Khon Sawan, Chaiaphum Province, using a research and development approach. The manuscript was well prepared. Some suggestions are shown below to improve the quality of manuscript.
In Abstract, how the findings from this work guide the medical waste management of bed-bound patients in Thailand should be briefly described at the end of Abstract.
I rewrite and added the findings briefly described at the end of the abstract.
The purpose of this study was to encourage innovative participation in the management of medical waste by bedridden patients in the research region of Khon Sawan, Chaiaphum Province, through research and development. The steps are as follows: Phase 1: A study of bedridden patient waste management situations. using the amount of waste generated as an innovation with relatives, non-relatives, village health volunteers (VHVs), and community leaders. Phase 2: Developing creative waste management engagement requires two steps: 1) analyzing the problem or its cause and generating management alternatives through collaborative brainstorming with a community member, and 2) gathering the thoughts and suggestions of a number of agency specialists. The outcome is a novel model of participation in waste management by bedridden patients termed "4 Joins of Power," which includes 1) participatory activities and enhancing community knowledge and attitudes, and 2) providing information on the management of each type of waste. 3) cooperation in waste management (analytical thinking, planning, execution, etc.) and are regulated by mutually agreed-upon rules. 4) expanding the waste management network jointly. Phase 3 is the innovation trial, while Phase 4 is the innovation assessment. Using the paired t-test to compare pre-and post-development knowledge and attitudes, and conducting qualitative data analysis. In Phase 3, after implementing the collaborative innovations, the average knowledge(=13.23) and attitudes(=4.14) regarding waste management increased considerably (P<.05), and in Phase 4, waste management behavior comprising sorting, storage, and disposal was observed. There were progressively substantial gains (=4.25 and=4.27, respectively). 93.50% of the most collaborative individuals were satisfied in line 11-30.
Briefly described at the end of Abstract.
In order to reduce the amount of waste that must be sorted and collected, it is necessary to emphasize from the commencement the participation of people and networks from all sectors in the area in joint thinking, planning, and comprehensive analysis ensure the sustainability of waste management in the community in line 30-33.
Section “2. Population” and section “3. Step of the research process” should be combined into one section such as “project design”. Some description such as “Data analysis” should be combined. Moreover, figure 1 should be described in “project design” rather than section “Introduction”.
I revised the methodology of the study as well as the classifications of the materials and the analysis (data collecting technique, population/sample, data collection instrument, and statistics), and I increased the lengths of the study for each phase from lines 240–391.
“4. The result” should be revised “4. Results”.
I changed lines 401.
In the manuscript, the description “3.1 Depending ……”, “3.2 Making home ……” should be revised because it easily misleads readers as sub-section of section “3.”.
I added sub-section as
“3.1) Depending on the type of waste, residential waste and medical waste from bedridden patients are separated. The output of waste sorting actions is the ability to source-separate waste. They sorted hazardous and disease-spreading waste from general waste and collected infectious waste from bedridden patients.
3.2) By encouraging knowledge and skills about separation and the use of organic waste to manufacture biofertilizers at home, organic waste was segregated from other wastes and preserved from decay. In addition, the advantages of household bio-fertilizers may be exploited in agriculture.”
There is less information in Table 1- table 4, some tables should be combined or should be described in words rather than in table.
Answer: I changed and rewrote the description in Table 1 in lines 511-515, table 2 in line 520-524, table 3 in line 538-542, and table 4 in line 545-548.
At the end of section “5. Discussion”, brief discussion of how the findings from this work guide the medical waste management of bed-bound patients in Thailand should be added.
In the discussion of the results, the study team has divided the success elements for adopting the "4 Joins of Power" into four distinct categories: 1. Social support 2. Local policy 3. Social engagement, and 4. sharing and learning in lines 557-705.
brief discussion
I added title
“The novelty of it.
The invention "Four Joints of Power" derived from the study may be regarded as a novelty since it introduces new concepts and behaviors. It has been demonstrated and developed as a process according to the study (Phase 1-4) and as a systematic approach using data on the number of infectious wastes in the community, which is empirical data, as the starting point for data analysis and with various stakeholders to participate in problem-solving, analyzing social engagement, and developing learning about infectious waste management. Later, it was demonstrated by putting innovation to the proof and reviewing its usage, starting with the categorization of waste categories and the collection of attitudes and levels of satisfaction about the innovation's application in lines 753-762.
Some references are suggested to support the description of waste and pollutant removal: A review of disposable facemasks during the COVID-19 pandemic: A focus on microplastics release, Chemosphere, 2022, 137178; MXenes as Heterogeneous Fenton-like Catalysts for Removal of Organic Pollutants: A Review, Journal of Environmental Chemical Engineering, 2022, 108954; Metal-organic frameworks-derived catalysts for contaminant degradation in persulfate-based advanced oxidation processes, Journal of Cleaner Production, 2022, 134118; A review on persulfates activation by functional biochar for organic contaminants removal: Synthesis, characterizations, radical determination, and mechanism, Journal of Environmental Chemical Engineering 2021, 9 (5), 106267.
I added these references in lines 745-752.
The Conclusions section is redundant and should be refined.
I rewrite in line 707-744.
Round 2
Reviewer 4 Report (New Reviewer)
The paper has been improved but still has issues:
1. The English language and writing style need to be significantly improved. The paper should be reviewed and corrected by a native speaker.
2. Lines 188-191 are not clear.
3. Line 200 stating 0.41 kg of waste is generated daily in Thailand can’t be correct.
4. Lines 216-218 what are these percentages for? In 8.5% of what?
5. Line 241 “in order” mentioned twice which doesn’t read well.
6. Line 336 January 2021 can’t be correct.
7. Line 356 “improvement. 0.05 levels” doesn’t seem right.
8. The section Result should be Results
9. Line 743 the statement that the problem is when gloves are releasing “microplastics in the atmosphere” is wrong. The problem with microplastics is when they are released in water, microplastics in atmosphere is not a problem as they deposit very fast.
10. The next sentence (lines 744-747) has nothing to do with the presented work so it doesn’t belong to Conclusions section.
11. “The novelty of it.” as a sentence is not right.
Overall the language of the paper needs a significant rework.
Author Response
Dear Reviewer
I am really appreciative of the research team's insightful comments and the helpful advice they gave to include in the manuscript in order to enhance the overall quality of the article and make it more interesting and beneficial to the audience.
Best regard
Dr. Chaiwong
Suggestion
The paper has been improved but still has issues:
1.The English language and writing style need to be significantly improved. The paper should be reviewed and corrected by a native speaker.
I did
- Lines 188-191 are not clear.
Old sentence: Although, the quantity of infectious waste produced is less as compared to the overall healthcare waste, the poor waste management practices by healthcare workers mix this waste with non-infectious waste and contaminate the whole lot as infectious waste.
New sentence: Even if the amount of infectious waste produced is negligible in proportion to overall medical waste, these wastes will arise from waste management techniques that are not in compliance with the hygienic principles of infectious waste management. When combined with other debris, this can constitute an infectious waste in lines 145-148.
- Line 200 stating 0.41 kg of waste is generated daily in Thailand can’t be correct.
I apologized for to mistake with the reference. I added a new reference in lines 55-57.
- Lines 216-218 what are these percentages for? In 8.5% of what?
and infectious waste as Hepatitis B Surface Antigen (HBsAg) positivity in 8.5%
Answer: I mean that 8.5% was Hepatitis B Surface Antigen (HBsAg) positivity.
- Line 241 “in order” mentioned twice which doesn’t read well.
Old sentence: Figure 1 provides an overview of the stages that are involved in the process of developing innovation for each activity that occurs throughout phases 1-4. These activities are carried out in order to generate innovation in order to accomplish the "4 Joints of Power."
New sentence: Figure 1 provides an overview of the stages involved in the process of developing in-novation for activities in phases 1-4. This is done to encourage new ideas named "4 Joints of Power." In line 157-158.
- Line 336 January 2021 can’t be correct.
I changed “The concept was evaluated from September 2021 to January 2022” to line 234
- Line 356 “improvement. 0.05 levels” doesn’t seem right.
Old sentence: Frequency, percentage, mean, and standard deviation were utilized to explore descriptive statistics, and a t-test was performed to analyze persons' knowledge and attitude toward medical waste management before and after significant improvement. 05 levels.
New sentence: Descriptive statistics such as frequency, percentage, mean, and standard deviation were used, and analytical statistics such as a t-test were used to compare individuals' knowledge and attitudes about medical waste management before and after at a statistical significance level of.05. in line 246-248.
- The section Result should be Results
I changed from “Result” to “Results” in line 270.
- Line 743 the statement that the problem is when gloves are releasing “microplastics in the atmosphere” is wrong. The problem with microplastics is when they are released in water, microplastics in atmosphere is not a problem as they deposit very fast.
I deleted this sentence
- The next sentence (lines 744-747) has nothing to do with the presented work so it doesn’t belong to Conclusions section.
I deleted the sentence
“The creation of "4 Joints of Power" enables caretakers of bedridden patients to segregate the types of medical waste produced by bedridden patients in the community so that it may be disposed of in accordance with academic requirements. It will also reduce the environmental effect, particularly in the case of medical plastics such as gloves. that can release microplastics into the atmosphere. In the future, persulfate activation by functional biochar for the removal of organic contaminants, metal-organic framework-derived catalysts for contaminant degradation in persulfate-based advanced oxidation processes, and MXenes as heterogeneous fern-like catalysts for the removal of organic pollutants will become more pre-dominant [56-59]”.
- “The novelty of it.” as a sentence is not right.
I deleted the topic “The novelty of it “ but still
The "4 Joints of Power" innovation consists of 4 parts:1) community engagement; 2) providing knowledge on how to handle infectious waste in the community; and 3) community, workplace, and individual collaboration. 4) The network strength of social movements, academic institutions, and communities. Phase 3 is the outcome of investigations that have been proven and developed in line with Phases 1 and 2. The amount of infectious waste produced by bedridden patients serves as the starting point for data analysis, and for stakeholders to assess and address problems, as well as to acquire knowledge of infectious waste management. Phase 3 was effective for assessing stakeholder knowledge and attitudes. In Phase 4, waste classification behavior was evaluated to improve the precision of waste classification. Thus, the four Joints of Power can be considered innovations in the process of infectious waste management for immobile patients.
Overall the language of the paper needs a significant rework.
I sent it to a native speaker for proofreading.
Note: The study team revised the red text in order to make it simpler and more straightforward to comprehend.

Round 3
Reviewer 4 Report (New Reviewer)
Authors responded to reviewers comments.
This manuscript is a resubmission of an earlier submission. The following is a list of the peer review reports and author responses from that submission.
Round 1
Reviewer 1 Report
The article submitted by Pattra et al. “4 Joints’ innovation of community involvement in medical 1 waste management of bed-bound patients in Thailand” suggests methods to foster the waste separation and treatment of bed-bound patients medical waste in district Khon Sawan in Thailand. The idea and the publication of such methods that involve professional and non-professional persons in awareness for potentially dangerous waste and the necessity to treat that waste carefully very important, particularly in countries (or regions) with simple waste management capacities.
However, the manuscript lacks some important information. In addition, it is very hard to read and to understand, as it not structured well and has a considerable amount of typos in it. Furthermore, it fits only marginally the subject of “Sustainability”, at least in the sense of waste management.
Therefore, I suggest a major revision.
General issues:
The manuscript is too long as the same ideas and explanations are repeated throughout the manuscript.
If this study was made for a particular region, this region should be described intensively. How, does it compare to other regions with regard to waste and medical management. Also, the interview and tested persons should be described by their socio-economic factors and how experienced they are. Use tables and graphs for such a thing. And use a map to display where this region is what is the GDP compared to other regions etc.
The introduction is to long and should be point to the problem that the authors want to solve with their method for that particular region.
The discussion starts somehow like a summary or conclusion section and is also to long. Also the studies that the authors mention appear not in context with their results. It is read more like a review.
Generally, the authors should not use the terms rubbish, trash or garbage, but exclusively the term waste.
This may also be because the manuscript is hard to read, not exclusively but also caused by the careless writing, points are missing and letters are capitalized randomly.
Here are some more particular issues
L stands for line.
L38-40 Why was medical described this way, and how is it described today?
L41-44: Do the authors mean that medical waste is generally contaminated with 10-25% of infectious pathogens, or are they potentially contaminated and pose a risk? Such numbers appear very high and are also different by region.
L46: replace trash by waste.
L53 replace garbage by trash.
L56 according to which statistics?
L64-65 replace “rubbish my sweep” to components from the waste are leached out and enters rivers …
L73-75 the sentence is hard to read and to understand. Please reformulate.
L82: “use is expected rise” to instead of “will rise”.
L94: Why do the authors use a prognosis of the quantities for the year 2017? Today is 2022.
L2 92-95: In general, the sentence is not very well formulated as the authors use for numbers for one year which is 2017. It should be rephrased.
L95-97: Its difficult to understand the context of this comparison. Why would you compare Thailand to China? And why would you compare residual waste to medical waste.
L98-99: what do you mean with sharp workplace injuries. Do you mean injuries with sharps at the workplace.
L99-101: Did the dustmen have a higher prevalence of infections than a comparable group of workers that do not work with waste.
Figure one has bad proportions and resolution
L220 -229: These chapters sound like is in protocol language.
L499: “The conclusions of the study might be discussed as follows” This sounds as if the authors also do not know how to discuss their study.
L536: “Based on research done in Iran, it is vital to have a thorough awareness of the waste” you should put this study or case in context to your own case study.
L596 3.2 kgbed/ day the dimension has to be separated from the number.
L602 – 603 the sentence does not make sense.
Author Response
We appreciate the comments you provided about MS, which helped the author make improvements. Rewriting the statement in the highlighted portion in blue not only fixed the issue but also made it significantly better. The author has made some changes to the content that are highlighted in purple to make it more understandable.
Best regard

Reviewer 2 Report
Infectious waste as sharps pose and 41 diseases such as AIDS, hepatitis B, and C accounts for 10% to 25% of medical waste, 42 whereas non-infectious waste accounts for 75% to 95% (Askarian et al., 2010) (Graikos et 43 al., 2010). - unclear
75–95% of certain 56 biological waste is safe, whereas 10–25% is dangerous - do not start the statement with a figure
line 117 - the researcher but there are 3 researchers
Lines 120-124 - very unclear as to the objectives of the study. Is it an investigation or an invention?
Objectives were to investigate the current state of waste disposal for bedridden, create ideas, and evaluate bedrid- 126 den patients' engagement in waste management. - whose point of view is this?
line 129 - what is the meaning of investigation in the context of the objective?
what is your definition of bedridden patient?
why is the focus on this kind of patient?
how did the caretakers estimate the amount of garbage? what are the sicknesses of the patients?
How did they analyze the text data they gathered from interviews?
453 - how did you measure participant knowledge?
what are the bases of the recommendations?
Author Response
We appreciate the comments you provided about MS, which helped the author make improvements. Rewriting the statement in the highlighted portion in green not only fixed the issue but also made it significantly better. The author has made some changes to the content that are highlighted in purple to make it more understandable.

Reviewer 3 Report
This paper gives an investigation on the innovation of community involvement in medical waste management of bed-bound patients in Thailand. Overall, this topic is interesting, and this paper can be considered after revision.
(1) The abstract is too long, which should be shorten below 250 words.
(2) The references in text is not corresponding to the references list. The checked reference should be listed as "[1][2][3]…" rather than the current cited form.
(3) The author should give a general introduction on the various waste worldwide, including the construction waste, food waste and medical waste, which is closely related to the sustainability topic. The following one reference may be helpfully “Characterization of sustainable mortar containing high-quality recycled manufactured sand crushed from recycled coarse aggregate” (https://doi.org/10.1016/j.cemconcomp.2022.104629)
(4) The conclusion should be listed one by one.
(5) The author should give a comparison on the findings of this work and previous investigations, and the shortage should this work should be further highlighted at the end for this paper.
Author Response
We appreciate the comments you provided about MS, which helped the author make improvements. Rewriting the statement in the highlighted portion in yellow not only fixed the issue but also made it significantly better. The author has made some changes to the content that are highlighted in purple to make it more understandable.

Round 2
Reviewer 1 Report
The manuscript is now more sound and can be considered for publication.
I strongly recommend to change more the often use everday terms like "rubbish", "garbage" etc. to more scientific terms like "waste". This can be done during the proof reading process.
Author Response
Dear Reviewers
"Four Joints of Power: Innovation of Community Involvement in Medical Waste Management of Bed-Bound Patients in Thailand" is the title of the study that was conducted. The identification number of the manuscript is as follows: 2076660 (1992059). I followed the reviewer's proposal for the new manuscript, which was to prove, modify, and rewrite the content that the reviewer had provided. In addition, I have improved some of those phrases by revising some of them to make them more understandable, which has resulted in MS having better importance to your journal being of greater value to your journal.
Finally, all authors have approved the manuscript and agreed to publication.
Yours Sincerely
Sanhawat Chaiwong, PhD
